# Acidified Nitrite Accelerates Wound Healing in Type 2 Diabetic Male Rats: A Histological and Stereological Evaluation

**DOI:** 10.3390/molecules26071872

**Published:** 2021-03-26

**Authors:** Hamideh Afzali, Mohammad Khaksari, Sajad Jeddi, Khosrow Kashfi, Mohammad-Amin Abdollahifar, Asghar Ghasemi

**Affiliations:** 1Endocrinology and Metabolism Research, and Physiology Research Centers, Kerman University of Medical Sciences, Kerman 7616913555, Iran; hamide_afzali@yahoo.com (H.A.); mkhaksari@kmu.ac.ir (M.K.); 2Endocrine Physiology Research Center, Research Institute for Endocrine Sciences, Shahid Beheshti University of Medical Sciences, Tehran 1985717413, Iran; Sajad.jeddi@sbmu.ac.ir; 3Department of Molecular, Cellular and Biomedical Sciences, Sophie Davis School of Biomedical Education, City University of New York School of Medicine, New York, NY 10031, USA; Kashfi@med.cuny.edu; 4Department of Biology and Anatomical Sciences, Faculty of Medicine, Shahid Beheshti University of Medical Sciences, Tehran 1985717413, Iran

**Keywords:** acidified nitrite, diabetic wound, histology, nitric oxide, type 2 diabetes, VEGF, hydroxyproline

## Abstract

Impaired skin nitric oxide production contributes to delayed wound healing in type 2 diabetes (T2D). This study aims to determine improved wound healing mechanisms by acidified nitrite (AN) in rats with T2D. Wistar rats were assigned to four subgroups: Untreated control, AN-treated control, untreated diabetes, and AN-treated diabetes. AN was applied daily from day 3 to day 28 after wounding. On days 3, 7, 14, 21, and 28, the wound levels of vascular endothelial growth factor (VEGF) were measured, and histological and stereological evaluations were performed. AN in diabetic rats increased the numerical density of basal cells (1070 ± 15.2 vs. 936.6 ± 37.5/mm^3^) and epidermal thickness (58.5 ± 3.5 vs. 44.3 ± 3.4 μm) (all *p* < 0.05); The dermis total volume and numerical density of fibroblasts at days 14, 21, and 28 were also higher (all *p* < 0.05). The VEGF levels were increased in the treated diabetic wounds at days 7 and 14, as was the total volume of fibrous tissue and hydroxyproline content at days 14 and 21 (all *p* < 0.05). AN improved diabetic wound healing by accelerating the dermis reconstruction, neovascularization, and collagen deposition.

## 1. Introduction

Worldwide, diabetes prevalence is currently 9.3% among adults and is estimated to reach 10.9% in 2045 [1]. Diabetic foot ulcer (DFU) is one of the most disabling complications of diabetes [2], afflicting 6.3% of diabetic patients worldwide [3]. The incidence of DFU is increasing, with a lifetime incidence that can be as high as 25% in diabetic patients [2]. DFU frequently results in amputation [2], and compared with nondiabetic subjects, the rate of lower limb amputation is 10–33 times higher in diabetic patients [4,5]. In addition, the risk of mortality is 50% higher in diabetic patients who have a history of DFU [6]. The treatment of DFU is a slow and challenging issue; according to a meta-analysis of randomized clinical trials, only about 30% of individuals with a neuropathic DFU will heal within 20 weeks of commencing standard care [7]. Moreover, even in superficial ulcers, the median time to healing exceeds two months [8]. This issue warrants further research to find new treatments for the diabetic wound.

The process of normal wound healing has overlapping and synchronized phases, including inflammation, proliferation, and tissue remodeling [9]. In the inflammation phase, nitric oxide (NO) leads to vasodilation [10], serves as a cytokine modulator [11], and has antiplatelet and antibacterial effects [12]. During the proliferative phase, NO promotes re-epithelialization [13] and stimulates angiogenesis and neovascularization [14]. In the remodeling phase, NO acts as an enhancer for collagen deposition [15] and reduces the expression of matrix metalloproteinases (MMPs), which cleave collagen, fibronectin, and other components of the extracellular matrix (ECM) [16]. The inhibition of endogenous NO synthesis in inducible NO synthase (iNOS) [17] and endothelial NO synthase (eNOS) [18] knockout mice leads to the thinning of granular tissue [17], delayed re-epithelialization (17), impaired angiogenesis (18), and, consequently, delayed wound closure [17,18]. In addition, a decreased NO bioavailability in type 2 diabetes (T2D) impairs wound healing by reducing the granular tissue formation and collagen deposition [19].

It has been shown that increasing the endogenous NO synthesis [20] and applying exogenous NO on the wound [11,21] accelerates wound closure and improves tissue quality in diabetic rats. A randomized controlled trial using a NO-generating dressing for treating DFU showed improved healing, 88.6% ulcer area reduction at 12 weeks, with only 46.9% in those treated with standard dressings [22]. Acidified nitrite, which produces NO, improves wound healing in T2D (11); however, this effect’s underlying mechanisms have not been fully understood. The favorable effects of nitrate/nitrite against T2D, including the regulation of glucose homeostasis, improvement of insulin resistance, and vascular function, have been reported in animal studies reviewed elsewhere [23,24,25]. These effects are related to the ability of nitrate/nitrite to be converted to NO [23]. We recently reported that acidified nitrite accelerates the wound-healing process in type 2 diabetic rats by restoring the delayed inflammatory response and augmenting the antioxidant defense mechanisms [11]. Beneficial effects of acidified nitrite have been reported in infected wounds in humans [26], burn wounds in normal mice [27] and rats [28], and, also, in the diabetic wounds in genetically diabetic (db/db) mice [29]. The ability of acidified nitrite to effectively and simply deliver NO makes it a potentially clinical available method for treating diabetic wounds. This study investigated the effects of acidified nitrite on re-epithelialization, the dermis structure, new blood vessel formation, and collagen deposition during the wound-healing process in type 2 diabetic rats using a histological and stereological approach.

## 2. Results

### 2.1. Changes in Body Weight

As shown in Figure 1, the body weights in control and diabetic rats were similar at the start of the study. Normal diet-fed rats (C and CN) had a gradual increase in their body weights from day 0 to day 56 of the study. The body weights of the untreated diabetic and acidified nitrite-treated diabetic rats (high-fat diet (HFD)-fed rats) were significantly higher compared to the untreated control rats at day 56 (329.2 ± 4.6 vs. 298.7 ± 7.6 g and 325.7 ± 8.5 vs. 298.7 ± 7.6 g, respectively, *p* < 0.001). The application of topical acidified nitrite did not affect the body weights.

### 2.2. Effect of Acidified Nitrite on Indices of Re-Epithelialization

#### 2.2.1. Numerical Density of Basal Cells

The numerical density of the basal cells, which are immature keratinocytes (Figure 2a), was significantly lower in the untreated diabetic rats compared to the untreated controls at all time points, except at day 3 after wounding (*p* < 0.05 for all). These data indicate a general decrease in re-epithelialization in the diabetic wound. The topical application of acidified nitrite significantly increased the numerical density of basal cells on day 28 after wounding in diabetic rats compared to the untreated ones (1070 ± 15.2 vs. 936.6 ± 37.5 basal cell/mm^3^, *p* = 0.025). However, it did not affect days 7, 14, and 21 after wounding.

#### 2.2.2. Epidermal Thickness

As shown in Figure 2b,c, on day 28, after wounding, the epidermal thickness in the untreated diabetic rats was significantly lower compared to the untreated controls (44.3 ± 3.4 vs. 55.1 ± 3.5 μm, *p* = 0.041); this is probably due to a decrease in the number of basal cells in this layer of the skin in the diabetic rats.

The application of acidified nitrite significantly increased the epidermal thickness in the diabetic rats compared to that of the untreated ones (58.5 ± 3.5 vs. 44.3 ± 3.4 μm, *p* = 0.009).

### 2.3. Effect of Acidified Nitrite on Indices of Dermis Structure

#### 2.3.1. Total Volumes of the Granular Tissue and Dermis

As shown in Figure 3a, at days 3 and 7 after wounding, the total volume of granular tissue was significantly lower in untreated diabetic rats compared to untreated controls (7.2 ± 0.9 vs. 12.2 ± 0.7 volume/mm^3^, *p* = 0.006 at day 3 and 11.3 ± 1.5 vs. 15.8 ± 0.9, *p* = 0.012 at day 7). On days 14, 21, and 28 after wounding, there was no significant difference in the granular tissue volume between groups. The acidified nitrite application significantly increased the total volume of the granular tissue in diabetic rats only at day 7 after wounding (14.9 ± 1.1 vs. 11.3 ± 1.5, *p* = 0.045, compared to untreated diabetes) (Figure 3a,c).

As shown in Figure 3b, the dermis total volume was significantly lower in the untreated diabetic rats compared to the untreated controls at all time points after wounding (all *p* < 0.05), except on day 3. Compared to the untreated diabetic rats, the acidified nitrite application significantly increased the dermis total volume at days 14, 21, and 28 after wounding in the diabetic rats (*p* < 0.05 for all).

#### 2.3.2. Numerical Density of Neutrophils, Macrophages, and Fibroblasts

As shown in Figure 4a,b, on day 3 after wounding, the numerical density of the neutrophils and macrophages was significantly lower in the untreated diabetic rats compared to the untreated controls (1433 ± 8.8 vs. 1650 ± 15.1 neutrophil/mm^3^, *p* = 0.001 and 65.3 ± 1.1 vs. 82 ± 1.7 macrophage/mm^3^, *p* = 0.001), suggesting a delay in the occurrence of the inflammatory phase. However, the values were significantly higher in the diabetic rats at all other time points, suggesting the inflammatory state’s persistence (all *p* < 0.001). Compared to the untreated diabetic rats, the acidified nitrite application significantly (all *p* < 0.05) decreased the numerical density of the neutrophils at days 14, 21, and 28 after wounding in the treated diabetic rats. Acidified nitrite significantly increased the macrophage numerical density in the diabetic rats only at day 7 (88.5 ± 1.1 vs. 80.1 ± 2.9 macrophage/mm^3^, *p* = 0.017). These data indicate that NO stimulates the recruitment of macrophages to the injured tissue.

The numerical density of fibroblasts (Figure 4c) was significantly lower in the untreated diabetic rats compared to the untreated controls at all time points except at day 3 after wounding (all *p* < 0.0001). The topical acidified nitrite application significantly increased the numerical density of the fibroblasts at days 7, 14, 21, and 28 after wounding in the diabetic rats, suggesting that NO has a positive effect on the fibroblast migration and proliferation. Compared to the untreated controls, control rats treated with acidified nitrite had a significantly higher density of fibroblasts on day 28 after wounding (all *p* < 0.05).

### 2.4. Effect of Acidified Nitrite on New Blood Vessel Formation (Neovascularization)

#### 2.4.1. Wound Levels of Vascular Endothelial Growth Factor (VEGF)

As shown in Figure 5, on day 3, after wounding, the VEGF protein wound levels were similar between the control and the diabetic groups before starting treatment. At days 7 and 14 after wounding, the VEGF levels were significantly lower in the untreated diabetic rats compared to the untreated controls (9.8 ± 1.7 vs. 23.7 ± 3.4 pg/mg protein, *p* = 0.049 at day 7 and 18.3 ± 3.7 vs. 35.4 ± 6.4 pg/mg protein, *p* = 0.032 at day 14). On day 21, after wounding, a sharp increase in this marker was observed in the untreated diabetic rats. On day 28, after wounding, the VEGF protein level in the wound tissue was lower in the untreated diabetic rats compared to the untreated controls, although it was only marginally significant (*p* = 0.088).

Acidified nitrite application significantly increased the tissue levels of VEGF in the diabetic rats compared to the nontreated diabetic ones at days 7 (36.8 ± 4.6 vs. 9.8 ± 1.7 pg/mg protein, *p* = 0.0009), 14 (34.4 ± 3.7 vs. 18.3 ± 3.7 pg/mg protein, *p* = 0.044), and 28 (36.5 ± 7.3 vs. 16.4 ± 3.5 pg/mg protein, *p* = 0.012) after wounding.

#### 2.4.2. Number of Blood Vessels

The number of blood vessels (Figure 6a–c) was significantly lower in the untreated diabetic rats than the untreated controls at day 21 after wounding (2.3 ± 0.3 vs. 5.7 ± 1.0 blood vessel/400× field, *p* = 0.004). The acidified nitrite application significantly increased the number of blood vessels in the control and diabetic rats compared to the untreated ones (9.5 ± 0.6 vs. 5.7 ± 1 blood vessel/400× field, *p* = 0.002 in control rats and 6.9 ± 0.8 vs. 2.3 ± 0.3 blood vessel/400× field, *p* = 0.0003 in diabetic rats).

As shown in Figure 6c, increased *neovascularization* was observed in the macroscopic skin images on day 21 after wounding in both the control and diabetic rats treated with acidified nitrite.

### 2.5. Effect of Topical Acidified Nitrite Application on Collagen Deposition

#### 2.5.1. Total Volume of fibrous Tissue

As shown in Figure 7a, the fibrous tissue volume was significantly lower in the untreated diabetic rats compared to the untreated controls at days 7, 14, and 21 after wounding (all *p* < 0.05). Acidified nitrite application significantly increased the volume of the fibrous tissue in the diabetic rats compared to the nontreated ones at days 14 (48.5 ± 7.2 vs. 30 ± 5.2, *p* = 0.048) and 21 (62.4 ± 4 vs. 43.2 ± 6.4, *p* = 0.045) after wounding. The application of acidified nitrite significantly increased the fibrous tissue volume in the control rats compared to the nontreated ones at day 14 after wounding (79.8 ± 2.2 vs. 60.4 ± 8.9, *p* = 0.019).

#### 2.5.2. Hydroxyproline Content

As shown in Figure 7b, the wound’s hydroxyproline content was significantly lower in the untreated diabetic rats than the untreated controls at days 14 and 21 after wounding (*p* < 0.05). The application of acidified nitrite significantly increased the hydroxyproline content in the diabetic rats compared to the nontreated ones at days 14 (2.4 ± 0.1 vs. 1.4 ± 0.1 µg/mg protein, *p* < 0.0001) and 21 (2.2 ± 0.2 vs. 1.7 ± 0.1 µg/mg protein, *p* = 0.021) after wounding. Additionally, thee acidified nitrite application significantly increased the hydroxyproline content of the control rats compared to the nontreated ones only at day 14 after wounding (3.2 ± 0.1 vs. 2.7 ± 0.1 µg/mg protein, *p* < 0.0001).

Figure 7c shows Masson’s trichrome-stained sections in the different groups; the deeper stained blue color in the trichrome-stained sections indicates an increased collagen deposition within the wound bed. The collagen deposition was found to be more condensed in the diabetic wounds treated with acidified nitrite, especially at days 14 and 21 after wounding.

## 3. Discussion

This study showed that the favorable effects of acidified nitrite on wound healing in type 2 diabetic rats are at least in part due to a rapid reconstruction of the dermis, augmentation of *neovascularization*, and acceleration of collagen deposition. This study documents for the first time that acidified nitrite as a NO-based therapy may have clinical relevance for the management of diabetic wounds. It increased the VEGF production and the number of fibroblasts while decreasing the neutrophil number during type 2 diabetic wound healing.

NO-based therapy is a potential candidate for diabetic wound healing [30]. The current study is a continuation of our previous report [11], where we showed that the time taken for 50% closure of a wound (CT50%) was higher in a diabetic wound compared to nondiabetic rats. The topical application of acidified nitrite decreased CT50% of wound healing (5.1 vs. 8.0 days, *p* < 0.001) in type 2 diabetic rats. In addition, on day 21 after wounding, the wounds closed in only two of the six untreated diabetic rats, while it was completely closed in all of the control rats. Acidified nitrite improved the closure time in diabetic rats by restoring the delayed inflammatory response and through augmentation of the antioxidant defense mechanisms [11].

To assess the acidified nitrite effects on wound re-epithelialization in diabetic rats, we measured the basal cell numerical density and the epidermal thickness in the wound samples. Type 2 diabetic rats had a lower numerical density of the basal cells from day 7 onwards and showed epidermal thinning at day 28 after wounding. Acidified nitrite increased both the number of the basal cells and the epidermal thickness at day 28 after wounding in rats with T2D. We did not find any report addressing changes in the number of basal cells or their alteration following NO administration in a diabetic wound. However, most basal cells are immature keratinocytes [31], activate upon acute skin injury, and migrate from the surrounding wound margins towards the center during the proliferation phase to reconstruct the epidermis through a process termed re-epithelialization [31]. In line with our results, a decreased epidermal thickness has been reported in the skin of type 2 diabetic rats [32]. It can also be due to a reduced number of basal cells, as reported in type 1 diabetic mice [33]. In addition, NO-producing probiotic patches have been shown to improve the epidermal maturity in ischemic wounds in rabbits [34]. Acidified nitrite did not affect the numerical density of the basal cells until day 28 after wounding. Still, it increased both the numerical density of basal cells and the epidermal thickness at day 28 after wounding, indicating that it can improve the skin’s epidermal integrity after wound healing.

This study assessed the effects of acidified nitrite on the dermis structure in the diabetic wound; the total volume of the granular tissue and dermis and numerical density of the neutrophils, macrophages, and fibroblasts were measured. The total volume of the granular tissue was lower on days 3 and 7 after wounding in rats with T2D, and acidified nitrite increased it on day 7 after wounding. Similar to our results, a delay in granular tissue formation has been reported in skin wounds of type 2 diabetic mice [35,36] and type 1 diabetic rats [37]. This delay may be due to the slow recruitment of inflammatory cells, decreased fibroblasts proliferation, decreased microvessel formation, and decreased collagen deposition, which are the tissue’s main granular components [38]. In line with our results, dressing of the wounds with polyvinyl alcohol (PVA)–NO hydrogels increased the granular tissue thickness in type 2 diabetic mice on day 8 after wounding [19]. In addition, iNOS-KO mice have a thinner thickness of the granular tissue at day 6 after wounding [17], indicating the necessity of NO for granular tissue formation.

Regarding the total volume of the dermis, it was significantly lower in the diabetic wound, and acidified nitrite increased it at days 14, 21, and 28 after wounding. Similar to our results, a thinner dermis has been reported in db/db mice [39]. Since the dermis is mainly composed of collagen and fibroblasts, which produce collagen [40], a decrease in the dermis’ total volume may be due to a decrease in the numerical density of the fibroblasts and the dermal collagen content, as shown in our study. We observed the fibroblast numerical density to be significantly lower in the diabetic wound, and acidified nitrite increased it at all time points. Similarly, a decrease in the number of fibroblasts has been reported in type 1 diabetic rats [41]. In addition, in human cultured fibroblasts, NO-releasing nanoparticles accelerated the fibroblast migration [15]. In our study, rats with T2D had a lower volume of fibrous tissue and hydroxyproline content, which are indicators of collagen deposition. On days 14 and 21 after wounding, acidified nitrite increased both these parameters. Similarly, a decrease in collagen formation in the wound bed has been reported in both type 1 and type 2 diabetic rats at days 14 and 21 after wounding [42]. Decreased collagen formation in a diabetic wound may be due to an overexpression of MMPs, which degrade ECM proteins, including collagen, fibronectin, elastin, and decrease expression of the tissue inhibitors of MMPs (TIMPs) [43,44,45]. It has been reported that S-nitroso-*N*-acetylpenicillamine (SNAP) and S-Nitrosoglutathione (SNOG), as NO donors, reduce the MMP-8 and MMP-9 mRNA expression in cultured human diabetic fibroblasts [16]. Molsidomine, an NO donor, increased the hydroxyproline content of the wound fluid in type 1 diabetic rats on day 10 after wounding [46].

The number of neutrophils and macrophages in the diabetic wound was lower on day 3 but higher on other days. Similar to our results, a delay in the initiation of an inflammatory response in the early stages (day 3 after wounding) and persistent inflammatory response in the late stages (days 7, 14, 21, and 28 after wounding) of the healing process have been reported in the excisional wound of type 2 diabetic mice [47]. In our study, acidified nitrite decreased the number of neutrophils at all time points in the diabetic wound. In line with these results, diethylenetriamine NONOate, an NO donor, increased the NO-mediated apoptosis of human-cultured neutrophils [48]. In our study, acidified nitrite increased the number of macrophages in the diabetic wound at day 7 after wounding. Later in the wound-healing process, acidified nitrite did not affect the macrophages. We did not find any reports addressing the effects of NO-based therapies on the number of neutrophils and macrophages in the diabetic wound. However, our findings were similar to those reported in normal BALB/c mice [15], where the topical application of an NO-releasing nanoparticle decreased the neutrophils and increased the macrophages in the wound bed on day 7 after wounding. A decrease in macrophage infiltration within the wound bed was shown in iNOS knockout mice on days 6 and 8 after wounding, and it was reported that iNOS-derived NO stimulated the recruitment of macrophages to damaged tissue. Acidified nitrite increases the iNOS protein levels and the concentrations of NO metabolites (nitrate+nitrite or NOx) in the wound tissues of diabetic rats on day 7 after wounding. This may explain the increase in wound macrophages in the acidified nitrite-treated diabetic rats at day 7 after wounding.

Our results indicate that acidified nitrite increases new blood vessel formation (neovascularization) within the wound tissues of rats with T2D, as confirmed by an increase in VEGF production and the number of blood vessels in the wound bed; the time course of VEGF protein production in the wound during the healing process was different between the diabetic and control rats. Diabetic rats had lower VEGF levels in the wound tissue at days 7 and 14 after wounding, a finding similar to that reported in type 2 diabetic rats [36]. In control rats, the VEGF production peaked on day 14 after wounding, whereas, in the diabetic rats, this peak was observed with a delay on day 21 after wounding. Acidified nitrite restored the wound VEGF level values to near the control levels. Increased wound VEGF levels in acidified nitrite-treated diabetic rats may be due to an increase in NO levels, because NO acts as a potent inducer of VEGF expression [49]. In support of this notion, S-nitroso-glutathione increased the mRNA and protein expressions of VEGF in human keratinocytes [49], and *L*-arginine increased the wound fluid VEFG levels in type 1 diabetic rats at day 5 after wounding [50]. In addition, iNOS inhibition decreased VEGF mRNA expression during wound healing in normal mice [49]. We recently reported that acidified nitrite increases the wound levels of NOx at days 7 and 14 after wounding in T2D rats [11], and in the current study with acidified nitrite-treated diabetic rats, there was a positive correlation between the wound tissue concentrations of VEGF and NOx_,_ particularly on day 14 after wounding (r = 0.444, *p* = 0.029; data not shown.

In our study, the acidified nitrite treatment increased the number of vessels at day 21 after wounding. Positive effects of NO therapy on new blood vessel formation in diabetic wounds have previously been shown [51,52]. In eNOS gene knockout (KO) mice, angiogenesis is lower, which impairs wound closure [18]. Cutaneous eNOS expression decreases in type 1 diabetic rats and causes impaired wound angiogenesis and healing [21,53]. NO plays an important role in endothelial proliferation, migration, and endothelial progenitor cell mobilization from the bone marrow to the circulation [54,55,56]. Wound angiogenesis is a key factor for successful wound healing, and therefore, NOS/NO signaling has potential clinical relevance for developing therapeutic targets in diabetic wound healing.

As a strength, the animal model of T2D used in this study (a combination of high-fat diet and low dose of streptozotocin) is closely related to the disease’s core pathophysiology in humans with hallmarks of insulin resistance and dysfunction of pancreatic β cells [57]. This study has some limitations; first, ulcers were created on the rats’ dorsum, whereas, in humans, ulcers mostly occur on the foot’s plantar surface [58]. However, self-licking in rats interferes with healing, because saliva contains growth factors and antimicrobial agents [59]. Second, we did not measure the effects of acidified nitrite on the basal cell migration, which is an index of re-epithelialization and is necessary for wound healing; in fact, NO promotes epidermal stem cells migration from the basal layer and hair follicles to the wound area [60]. Finally, we used a wound contraction model in rats, in which contraction of the panniculus carnosus muscle contributes to the wound-healing process; this muscle does not exist in humans, where wound repair is mainly driven by granulation and re-epithelialization [61]. Therefore, the extrapolation of this data to humans should be done with caution. In addition, contraction of the panniculus carnosus muscle can potentially affect measurements of the dermis volume, but because in our study, the total volume of the dermis in the diabetic rats was lower than that of the controls at all time points, the contraction probably had no significant effect on the measurement of the dermis volume.

## 4. Materials and Methods

### 4.1. Materials

Streptozotocin (STZ) was purchased from Sigma Aldrich (Hamburg, Germany). Sodium nitrite (NaNO_2_) from Merck (Darmstadt, Germany). Citric acid monohydrate was obtained from Sigma Aldrich (Vienna, Austria). An aqueous cream base was purchased from Farabi Pharmaceutical Co. (Tehran, Iran). Protease inhibitor cocktail tablets were from Roche Co. (Mannheim, Germany*).*

### 4.2. Ethical Approval

All animal procedures and care were carried out according to the standards for care and use of animals and approved by the ethics committee of the Research Institute for Endocrine Sciences (IR.SBMU.ENDOCRINE.REC.1397.268), Shahid Beheshti University of Medical Sciences.

### 4.3. Animals and Diet

All animals were purchased from the Research Institute for Endocrine Sciences, Shahid Beheshti University of Medical Sciences, Tehran, Iran. Wistar and Sprague–Dawley rats are the two strains mostly used in wound studies, with no preference/advantage or disadvantage in using either strain [62]. In this study, we used Wistar rats, because the metabolic effects of a high-fat diet appear earlier and more frequently in Wistar rats [63], and, also, because of their availability.

Throughout the study, rats in the control groups were fed a standard pellet diet (5.7% lipids, 22.1% proteins, and 72.2% carbohydrates) with a total calorie value of ~3100 kcal/kg. In contrast, diabetic rats were fed a high-fat diet (HFD) (58.8% lipids, 14.2% proteins, and 27.0% carbohydrates) with a total calorie value of ~4900 kcal/kg, as we described previously [57].

### 4.4. Groups and Study Design

Rats in the control and diabetic groups (*n* = 54/group) were divided into four subgroups: Untreated control (C, *n* = 30), acidified nitrite-treated control (CN, *n* = 24), untreated diabetes (D, *n* = 30), and acidified nitrite-treated diabetes (DN, *n* = 24). Rats in the untreated subgroups were assessed at five time points (*n* = 6/time point), i.e., at days 3, 7, 14, 21, and 28 after wounding, and rats in the treated subgroups were assessed at four time points (*n* = 6/time point), i.e., at days 7, 14, 21, and 28 after wounding [11].

As shown in Figure 8, in order to induce type 2 diabetes, a low dose of STZ (35 mg/kg freshly dissolved in 0.1-mM citrate buffer, pH 4.5) was intraperitoneally injected after 21 days of HFD manipulation [64]; rats in the control group were injected with an equal volume of citrate buffer. One week after STZ injection (day 28 of the study), rats with serum glucose concentrations greater than 150 mg/dL and lower than 350 mg/dL were followed. On day 54, if the serum glucose concentrations remained in this range, rats were considered to have T2D [64]. On day 56, a full-thickness skin wound was made on the backs of the animals. Acidified nitrite was applied from day 3 until day 28 after wounding [11].

Measurement of the wound levels of the VEGF, as well as histological and stereological evaluations, were performed at each time point (days 3, 7, 14, 21, and 28 after wounding). Evaluation of the epidermal thickness was performed on day 28 after wounding, when the dermis and epidermis were completely reconstructed [65]. Evaluation of the number of blood vessels was done on day 21 after wounding. The wound level of hydroxyproline was measured at days 14 and 21 after wounding [66].

### 4.5. Wound Induction

At day 56 of the study, rats were anesthetized by intraperitoneal injection of ketamine (70 mg/kg) and xylazine (10 mg/kg), and their dorsal surface hairs at 2 cm below the ears were trimmed using an electric clipper, and the remaining hairs were epilated using cold wax [67]. The skin was disinfected with 70% ethanol. Then, using an 8-mm-diameter biopsy punch (Kai industries Co., Ltd., Gifu, Japan), one full-thickness excisional wound was made on the back of the rats (Appendix A) [67].

### 4.6. Preparation and Application of Acidified Nitrite

The acidified nitrite cream was comprised of two creams: sodium nitrite cream and citric acid cream, which were made separately and combined at the time of application at the wound site [11]. The sodium nitrite cream contained 3.0% (*w*/*v*) sodium nitrite in an aqueous cream base. The citric acid cream contained 4.5% (*w*/*v*) citric acid monohydrate in an aqueous cream base. In the CN and DN subgroups, an equal amount (~20 mg) of sodium nitrite and citric acid were applied once daily from day 3 until day 28 after wounding. This method of topical cream application has been previously reported [11].

### 4.7. Measurement of Serum Glucose Concentration

After a 12–14-h fast, on days 28 and 54 before wounding, blood samples were collected from the tail vein and centrifuged for 10 min at 5000× *g*. Measurements of serum the glucose concentrations were done using the glucose oxidase method using commercially available kits (Pars Azmoon Co., Tehran, Iran) and a chemistry autoanalyzer (Selectra E chemistry analyzer, Dieren, The Netherlands); the intra- and inter-assay coefficients of variations (CVs) were 2.1% and 3.4%, respectively.

### 4.8. Tissue Collection

At the end of the study, 6 rats from each time point were sacrificed at days 3, 7, 14, 21, and 28 after wounding, and the wound, along with 5 mm of healthy adjacent skin, was harvested and immediately divided into two parts. One part was crushed into a fine powder under liquid nitrogen and then homogenized and sonicated (20 S, 40 W) using an ice-cold phosphate buffer solution (10 mM, pH 7.4, 1-mL/500 mg tissue, 1:2, *w*/*v*) containing a protease inhibitor cocktail [11] and then stored at −80 °C for measurement of the VEGF and hydroxyproline. The second part of the tissue was placed in labeled histology cassettes containing biopsy pads (biopsy pads used to prevent skin samples from curling during fixation) and then fixed in 10% formalin for histological and stereological evaluations.

### 4.9. Histological and Stereological Evaluations

The formalin-fixed tissues were dehydrated by ethanol and embedded in paraffin perpendicularly to the anteroposterior axis of the wound. Serial coronal sections with 10-μm thickness (section interval = 100 µm) were made perpendicular to the surface of the wound using a rotary microtome (DS4055, Did Sabz Co., Urmia, Iran). Then, using a systematic uniform random sampling, 8–10 sections per each animal were selected. Sections were mounted on glass slides, deparaffinized, and stained with hematoxylin and eosin (H&E), as well as Masson’s trichrome, for histological and stereological evaluations. A blinded observer performed all counts and measurements.

### 4.10. Indices of Re-Epithelialization

#### 4.10.1. Estimation of the Numerical Density of Basal Cells

The stereological optical dissector method was used to determine the numerical density of basal cells at days 3, 7, 14, 21, and 28 after wounding [68]. Basal cells are cuboid in morphology in the basal (the innermost layer) of the epidermis [69]. Numerical density (Nv) was calculated using the following formula:(1)$$Nv=\frac{\sum Q}{h.\frac{a}{f}.\sum p}\times \frac{t}{BA}$$
where Σ*Q* = number of nuclei, *h* = height of the dissector, $\frac{a}{f}$ = frame area, Σ*p* = total number of the unbiased counting frame in all fields, *t* = section thickness measured in every field using the microcator, and *BA* = block advance of the microtome.

#### 4.10.2. Measurement of Epidermal Thickness

On day 28, after wounding, the epidermal thickness was measured in H&E-stained sections. For this, three random sections per tissue sample were selected, and an image was taken from the wound center of each section at 400x magnification using a microscope (Olympus, BX51, Tokyo, Japan) equipped with digital imaging accessories (Nikon, DS-Fi1, Tokyo, Japan) [32]. Then, 10 random measurements were made using the straight-line measuring tool of ImageJ software (National Institutes of Health, Bethesda, Maryland*,* USA) [32]. Epidermal thickness was defined as the vertical distance between the top of the cornified epithelium to the inferior of the basal cell layer [32].

### 4.11. Indices of Dermis Structure

#### 4.11.1. Estimation of the Volume of the Granular Tissue and Dermis

The stereological Cavalieri’s method was used to estimate the total volume of the granular tissue and dermis at days 3, 7, 14, 21, and 28 after wounding [70]. The dermis is located immediately below the epidermis (from the epidermis to panniculus), and the dermal wound gap is filled with granular tissue. Granulation tissue is composed of an extracellular matrix produced by inflammatory cells, granulocytes, macrophages, fibroblasts, and new microvessels formed by invading endothelial cells in a complex with collagen bundles that are usually produced to fill up the injured space and serves as a scaffold for keratinocyte migration [38]. The formula for calculation of the volume (V) of the granular and dermis tissues was
(2)$$V=\sum p.\frac{a}{p}.t$$
where Σ*p* = total number of points hitting the skin tissue sections, $\frac{a}{f}$ = area related to each point projected on the skin tissue, and *t* = sampled sections distance.

#### 4.11.2. Estimation of the Numerical Density of Neutrophils, Macrophages, and Fibroblasts

Neutrophils have a multilobed nucleus and are approximately 6–9 μm in diameter, whereas macrophages are larger and contain a nucleus that is spherical in shape [69]. Fibroblasts are spindle-shaped cells that are characterized by the basophilic cytoplasm under H&E staining [69]. The numerical density of neutrophils, macrophages, and fibroblasts was estimated at days 3, 7, 14, 21, and 28 after wounding according to the optical dissector method described in Equation (1).

### 4.12. Indices of New Blood Vessel Formation (Neovascularization)

#### 4.12.1. Measurement of VEGF

Concentrations of VEGF in the wound tissue were measured at days 3, 7, 14, 21, and 28 after wounding using a rat-specific ELISA kit (ZellBio GmbH, Berlin, Germany). The kit sensitivity was 12.5 ng/L, and the intra-assay CV was <10%. The VEGF concentrations were normalized to the amount of sample protein concentrations, which measured using the Bradford method [71].

#### 4.12.2. Counting the Number of Blood Vessels

The number of blood vessels in the center of the wound site was counted in two random 400x magnification fields of each section per rat under the light microscope [72]. This counting was done in H&E-stained sections of samples at day 21 after wounding.

### 4.13. Indices of Collagen Deposition

#### 4.13.1. Estimation of the Volume of the Fibrous Tissue

The major structural protein of fibrous tissues in the skin is collagen [40], which stains blue with Masson trichrome and is acidophilic when it is stained with H&E. Estimation of the volume of the fibrous tissue was performed according to the Cavalieri’s method described in the Section 4.11.1 at days 3, 7, 14, 21, and 28 after wounding.

#### 4.13.2. Measurement of Hydroxyproline Content

The concentration of hydroxyproline, as a major component of collagen and a marker of collagen biosynthesis [66], was measured in the wound tissue samples at days 14 and 21 after wounding using the colorimetric method/kit (Kiazist Co., Hamedan, Iran). The hydroxyproline concentrations were normalized to the amount of sample protein concentrations. The kit sensitivity was 4 µg/mL, and the intra-assay CV was 1.2%.

### 4.14. Statistical Analysis

Statistical analyses were performed using GraphPad Prism software, (Version 6, La Jolla, and San Diego, CA, USA). Unpaired *t*-test was used for analyzing the data of wound levels of VEGF; numerical density of the cells; and the volume of the granular, dermis, and fibrous tissue before application of acidified nitrite at day 3 after wounding between the untreated control and diabetic subgroups. Two-way mixed (between–within) analysis of variance (ANOVA), followed by the Fisher’s post-hoc test, was used for analyzing the data of the wound levels of VEGF; numerical density of the cells; the volume of the granular, dermis, and fibrous tissue; and the hydroxyproline content at days 7, 14, 21, and 28 after wounding. One-way ANOVA followed by Fisher’s post-hoc test was used for comparing the number of vessels and epidermal thickness. All data are presented as the mean ± SEM, and two-sided *p*-values < 0.05 were considered to be statistically significant.

## 5. Conclusions

In conclusion, this study’s findings showed that acidified nitrite accelerates wound healing in type 2 diabetic rats by rapid reconstruction of the dermis and the augmentation of neovascularization and acceleration of collagen deposition in wound tissues (Figure 9). Regarding the fact that wound-healing materials need to be applied conveniently in a clinical setting and not be too expensive [19], these findings have implications in future diabetic wound treatments.

## Figures and Tables

**Figure 1 molecules-26-01872-f001:**
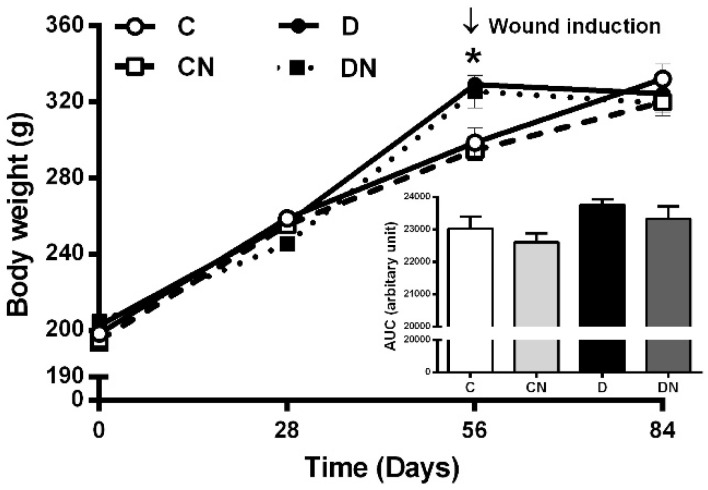
Changes in the body weights of rats during the study. Inset indicates the area under the curves. C, untreated control; CN, acidified nitrite-treated control; D, untreated diabetes; and DN, acidified nitrite-treated diabetes. * *p* < 0.001 compared to untreated control rats. Values are mean ± SEM, *n* = 6/subgroup.

**Figure 2 molecules-26-01872-f002:**
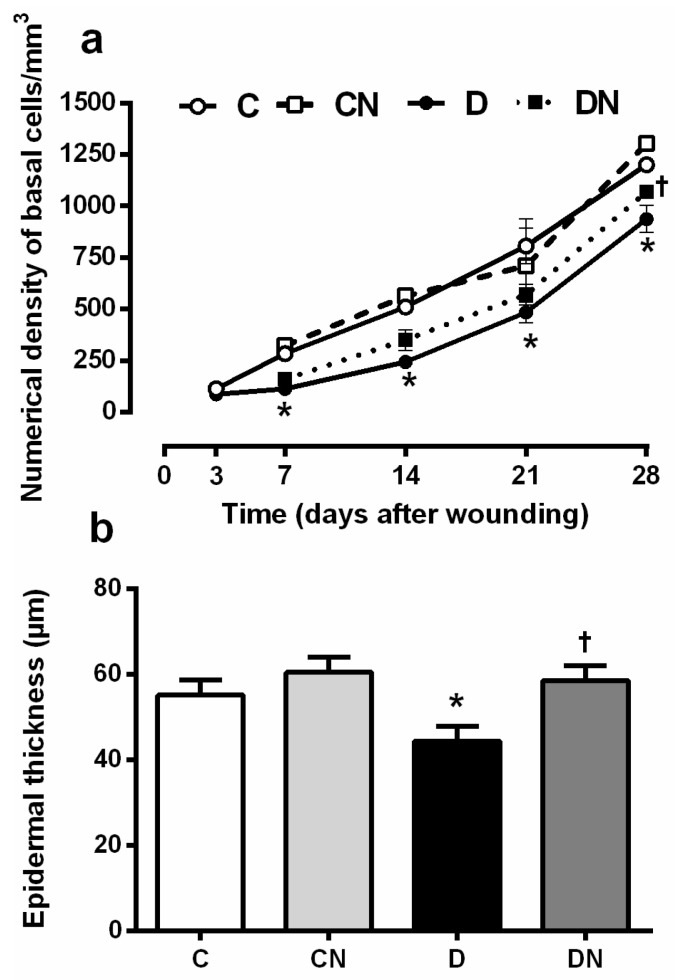
Effect of the topical application of acidified nitrite on the basal cell numerical density during wound healing (**a**) and epidermal thickness at day 28 after wounding (**b**). Representative images of hematoxylin and eosin (H&E)-stained sections at day 28 after wounding (400× magnification and scale bar = 50 µm) (**c**). E, Epidermis; V, microvessels; ▲, basal cells; C, untreated control; CN, acidified nitrite-treated control; D, untreated diabetes; and DN, acidified nitrite-treated diabetes. * *p* < 0.05 compared to untreated control rats, and †*p* < 0.05 compared to untreated diabetic rats. Values are mean ± SEM, *n* = 6/subgroup.

**Figure 3 molecules-26-01872-f003:**
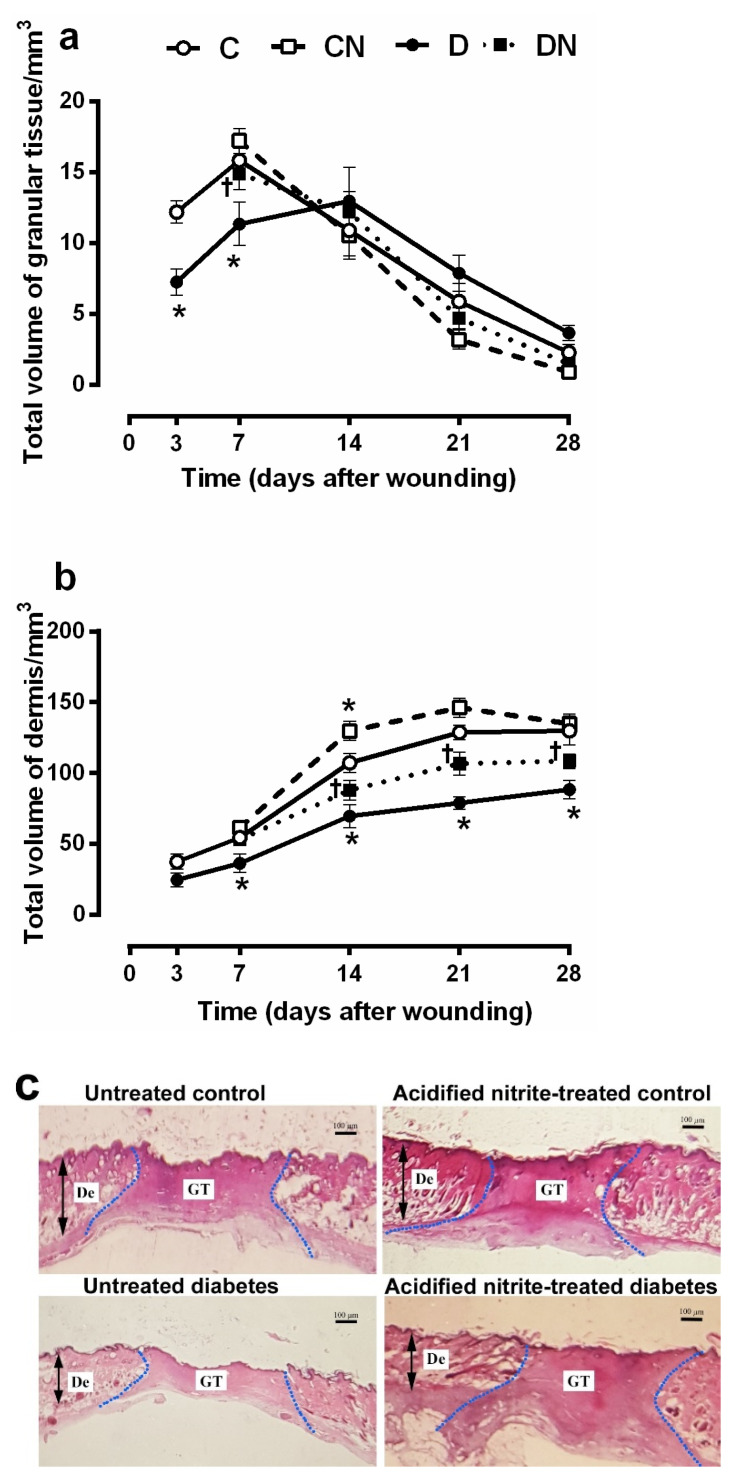
Effect of the topical acidified nitrite application on the total volume of the granular tissue (**a**) and the total volume of the dermis (**b**). Representative images of H&E-stained sections at day 7 after wounding (100× magnification and scale bar = 100 µm) (**c**). Dotted lines indicate the border between the granular tissue and dermis. GT, granular tissue; De, Dermis; C, untreated control; CN, acidified nitrite-treated control; D, untreated diabetes; and DN, acidified nitrite-treated diabetes. * *p* < 0.05 compared to untreated control rats, and † *p* < 0.05 compared to untreated diabetic rats. Values are mean ± SEM, *n* = 6/subgroup.

**Figure 4 molecules-26-01872-f004:**
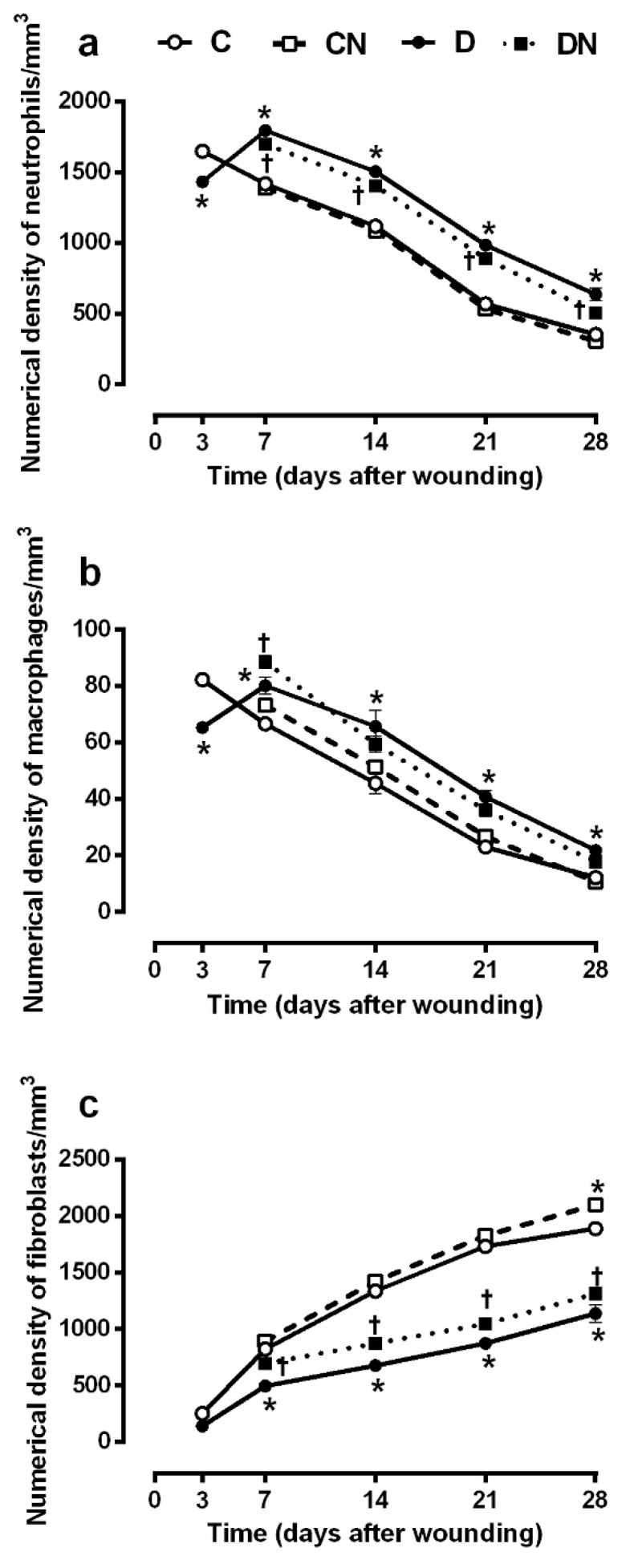
Effect of topical acidified nitrite application on the numerical density of neutrophils (**a**), macrophages (**b**), and fibroblasts (**c**). C, untreated control; CN, acidified nitrite-treated control; D, untreated diabetes; and DN, acidified nitrite-treated diabetes. * *p* < 0.05 compared to untreated control rats, and † *p* < 0.05 compared to untreated diabetic rats. Values are mean ± SEM, *n* = 6/subgroup.

**Figure 5 molecules-26-01872-f005:**
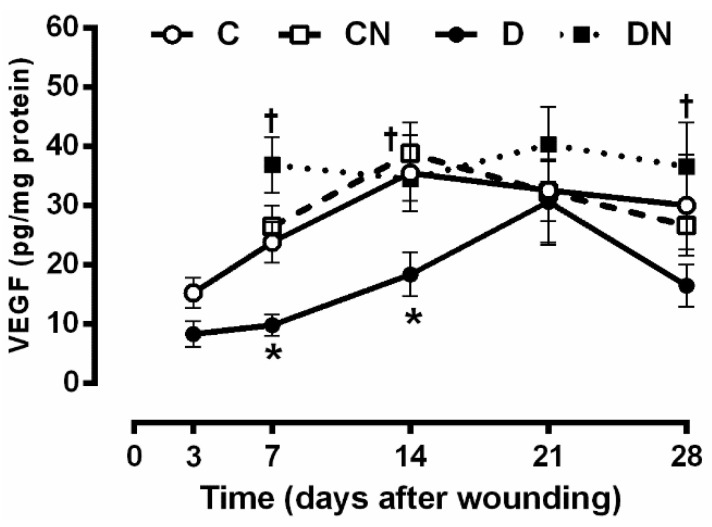
Effect of the topical acidified nitrite application on the wound tissue protein levels of the vascular endothelial growth factor (VEGF) (*n* = 6/subgroup). C, untreated control; CN, acidified nitrite-treated control; D, untreated diabetes; and DN, acidified nitrite-treated diabetes. * *p* < 0.05 compared to untreated control rats, and † *p* < 0.05 compared to untreated diabetic rats. Values are the mean ± SEM.

**Figure 6 molecules-26-01872-f006:**
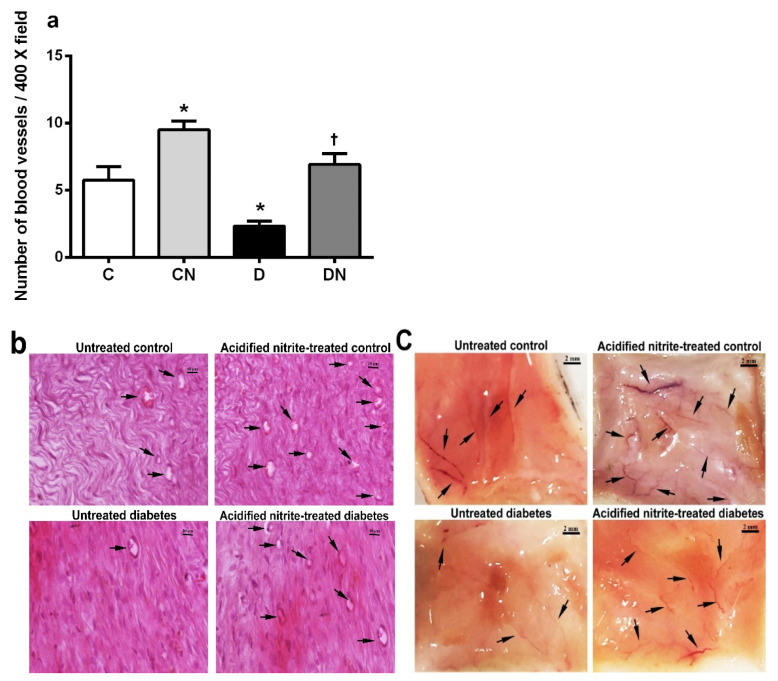
Effect of the topical acidified nitrite application on the number of blood vessels (*n* = 6/subgroup) (**a**). Representative images of H&E-stained sections at day 21 after wounding (400× magnification and scale bar = 10 µm) (**b**). Representative macroscopic images of skin at day 21 after wounding (scale bars = 2 mm) (**c**). Arrows, microvessels; C, untreated control; CN, acidified nitrite-treated control; D, untreated diabetes; and DN, acidified nitrite-treated diabetes. * *p* < 0.05 compared to untreated control rats, and † *p* < 0.05 compared to untreated diabetic rats. Values are the mean ± SEM.

**Figure 7 molecules-26-01872-f007:**
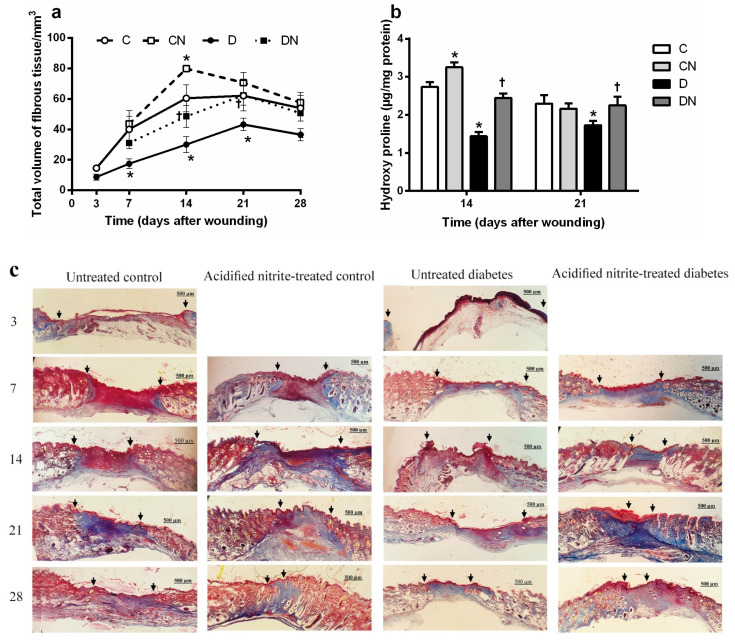
Effect of the topical acidified nitrite application on the total volume of fibrous tissue (**a**) and hydroxyproline content (**b**) (*n* = 6/subgroup). Representative images of Masson’s trichrome-stained sections (specific for collagen) at days 3, 7, 14, 21, and 28 after wounding (1000× magnification and scale bar = 500 µm) (**c**). Arrows, wound margin; C, untreated control; CN, acidified nitrite-treated control; D, untreated diabetes; and DN, acidified nitrite-treated diabetes. * *p* < 0.05 compared to untreated control rats, and † *p* < 0.05 compared to untreated diabetic rats. Values are the mean ± SEM.

**Figure 8 molecules-26-01872-f008:**
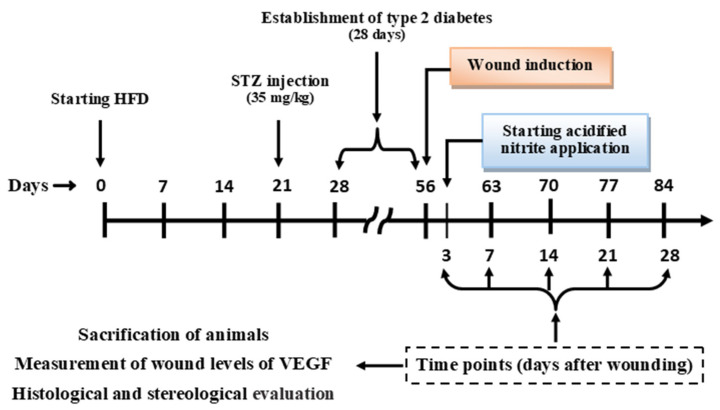
Study design. Evaluation of the epidermal thickness was performed on day 28 after wounding. Evaluation of the number of blood vessels was done on day 21 after wounding. Wound levels of hydroxyproline were measured at days 14 and 21 after wounding. Days 3, 7, 14, 21, and 28 after wounding were the time points after wounding (*n* = 6/time point). HFD, high-fat diet; STZ, streptozotocin; and VEGF, vascular endothelial growth factor.

**Figure 9 molecules-26-01872-f009:**
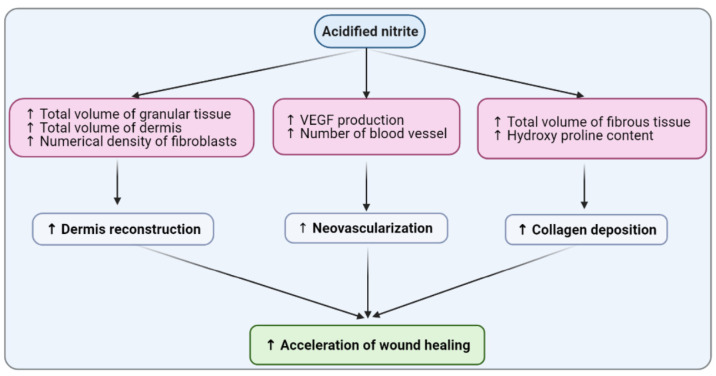
Mechanisms by which acidified nitrite improves diabetic wound healing.

## Data Availability

This study did not report any data.

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
