# Peer review of "Acidified Nitrite Accelerates Wound Healing in Type 2 Diabetic Male Rats: A Histological and Stereological Evaluation"

_molecules, 2021, doi:10.3390/molecules26071872_

Round 1
Reviewer 1 Report
The manuscript entitled “Effects of acidified nitrite on wound healing in type 2 diabetic male rats: A histological and stereological evaluation” demonstrated the wound healing mechanisms of re-epithelialization, dermis reconstruction, neovascularization, and collagen deposition mediated by Acidified Nitrate (AN) during wound healing in STZ-induced type 2 diabetic in experimentally-induced Wistar rats.
Overall, the objective of the study and presentation of the work is nice. The authors did a lot of work. However, similar work was previously published by the same authors in Nitric Oxide 103 (2020) 20–28. Therefore, this manuscript needed major revision by addressing the following questions.
Major comments –
Title
The title is wordy and passive (“Effects of….). It is suggested to modify [for example –ref # 11].
Introduction –
- Authors are suggested to mention why they chose Wistar rats in this study, although there are different types of rats are in common use for this type of study?
- MMPs play a key role in Collagen production. Is there any relation between these two? Some explanation in this regard in both Introduction and Discussion sections would further refine this manuscript. Therefore, it is suggested.
- Although the authors provided too many references pertaining to the literature, it is suggested to elaborate the importance of Nitric oxide (NO), AN, T2D and their inter-relation in this section. This helps to reach this work to wide range of readers.
Results
- In the manuscript result Section 2.1- 2.3, the author mentioned the day 7 increasing and day 21 or 28 decreasing. There is no clear explanation, for example, Epidermal thickness and density of basal cells of amount expression in untreated diabetes, acidified nitrite-treated diabetes
- It is advisable the proposed pathway mechanism for acidified nitrite-treated diabetes model (This would strengthen the importance of this study)
- Figure 3 (a-c) Day 7 numerical density of neutrophils, macrophages, and fibroblasts increased after day day21 decreasing. It is suggestible include the reason for after treatment acidified nitrite-treated diabetes
- Changes in body weight data need to include in the result part.
Discussion
- It is suggestable to add few lines with literature support about the novelty of this paper and discuss how it will reconstruct the dermis and new blood vessel formation.
Minor comments –
- From where the animals were purchased?
- What are the body weights of the animals (initial and later)? Is there any difference through-out the course of this study?
Figures
- Inside figure 2. (D) change to the De (Dermis)
- Figure 7c, day 7 (acidified nitrite-treated and acidified nitrite-treated diabetes) images are missing and correct the untreated control group day14 image size.
Reviewer 2 Report
The authors review the treatment of acidified nitrite on the healing of SZ-treated rat wounds. There are some significant flaws in the study that must be addressed:
1) This is a contraction model where the main healing is through shrinkage of the wound. There is no measure of wound size or time to complete closure. Is there an improvement in wound closure? That is the real clinically relevant question.
2) Some of the measures have no relevance to wound healing. The basal cell density at 28 days has no relevance. It appears that the wounds are healed (based on the histology). It would be more relevant to look at basal cell migration but not density. The epithelial thickness is really an indicator of how long the wound has been closed and has little relevance. Irritation of the wound can also cause thickening of the epithelium so it could be due to inflammation, not healing. The term for "granulation" tissue is incorrect - you use "granular". The thickness of granulation tissue and fibrosis could be a sign of wound closure or a sign of excessive scarring. The first is beneficial and the second is harmful. Hair follicles do not grow in the center of rodent contraction wounds. There may be hair follicles at the edge of the wound (from the normal skin) but it has no relevance to healing. I see no hair follicles within the wounds of any of the provided pictures. The volume of dermis may be smaller with greater contraction but still be healed. Failed healing may have more dermis so that value is questionable. The length of vascularization is not relevant - the density of vascularization is important.
3) A minor point - in Figure 4, the term is "sebaceous" gland, not "seabasses" gland (the latter refers to a type of fish).
Round 2
Reviewer 2 Report
The authors have addressed my issues.